# The Association of Insomnia and Stress on Cardiovascular Risk Factors during COVID-19 Confinement in the Mexican Population

**DOI:** 10.3390/ijerph20237135

**Published:** 2023-12-01

**Authors:** Sergio Urriza-Trejo, Héctor Hurtazo, Jorge Palacios, Martha Cruz-Soto

**Affiliations:** Escuela de Ciencias de la Salud, Campus Querétaro, Universidad del Valle de México, Boulevard Juriquilla No. 1000 A, Delegación Santa Rosa Jáuregui, Querétaro 76230, Mexico; dr.sergiourriza@gmail.com (S.U.-T.); hurtazo@gmail.com (H.H.); jorge.palaciosd@uvmnet.edu (J.P.)

**Keywords:** cardiovascular risk factors, insomnia, stress, COVID-19, pandemic

## Abstract

During the pandemic confinement, the WHO changed the term “social distancing” to “physical distancing”, to help people deal with the lack of social contact. As a result, there was an increase in mental health problems, including insomnia and stress, with a negative impact on cardiovascular health. The objective of this research was to identify the association between insomnia and stress and cardiovascular risk (CVR) during the pandemic in a sample of the general population in Mexico; the participants were chosen using the non-probabilistic method. The data were obtained from an online questionnaire about medical histories focused on cardiovascular risk, according to the Official Mexican Standards and Regulations for patients’ clinical records, NOM-004-SSA3-2012, along with an index for the severity of insomnia, measured with a seven-item guide, and an instrument to measure stress. The data were analyzed with descriptive statistics for several different variables: sociodemographics, stress, insomnia, and cardiovascular risk. Cardiovascular risk was compared to insomnia and stress variables, which led to statistically significant differences and correlations between the variables. Participants were divided into four groups with respect to CVR, from low to very high CVR. This research demonstrated that women were more susceptible to stress and cardiovascular risk. However, stress was a more major indicator of CVR than insomnia, but in the high and very high CVR groups, insomnia contributed along with stress; coping strategies reduced the risk in the high CVR group but did not function as expected with respect to reducing risk in the very high CVR group. These findings suggest that sleep patterns and mental health alterations present during the pandemic may persist even when the pandemic was declared as having ended and may contribute to increases in cardiovascular risk in the long-term.

## 1. Introduction

During the COVID-19 pandemic, the WHO changed the term “social distancing” to “physical distancing”, so that people might feel less isolated and lonely [1,2]. Isolation and loneliness are some of the consequences of the pandemic as shown in previous studies where individuals placed in quarantine reported a high prevalence of psychological anguish and even symptoms of psychological disorders, impacting mental health, with an increase in stress and insomnia in the whole population [3,4,5]. In addition to uncertainty about getting infected, when the pandemic will end, how many more deaths will follow, etc. [6,7,8,9], people experienced conditions such as burnout syndrome, as work or academic activities changed, due to personal stress, or due to any other adverse event; anxiety; and depression, all these leading to mental fatigue and resulting in an overall negative impact on mental and physical health [10]. Currently, some people are still presenting effects of long-COVID which involve different body systems including the cardiovascular and nervous systems, leading to physiological and mental health alterations [11]. There are different strategies to cope with stress, anxiety, and depression, which may reduce the mentioned health risks, but it is important that people get support and make sure they are followed up by a health care professional [12].

### 1.1. Cardiovascular Risk

The main cause of death worldwide is cardiovascular disease [1], and it has been shown that patients with this disease may have a higher susceptibility to COVID-19 infection; this has been observed [13] in patients presenting acute symptoms of COVID-19, causing a cytokine storm that represents a poor prognosis for patients infected with coronavirus [14].

Endothelial injury has also been reported [15], which produces arteriosclerosis, the main cause of cardiovascular disease. This leads to obstruction of the blood flow and increases the probability of vascular thrombosis and aneurism [15,16]. Cardiovascular risk factors (which can be nonmodifiable, modifiable, or metabolic) [17] have been shown to be present in patients prior to the contraction of COVID-19.

Nonmodifiable risks are age, for which the morbidity increases per decade lived [15]; gender [18]; family predisposition; and ethnic and racial groups [15,17]. Among the modifiable risks are tobacco use, which is one of the factors with the highest impact on cardiovascular disease [19,20,21]; undernourishment and lack of exercise, which were factors before the pandemic that have increased due to forced confinement; change in habits, such as working from home; increased stress; limited access to healthy and fresh food [6,22]; depression, which increases the risk of cardiac disease from two to five times [23,24]; and anxiety related to arteriosclerosis damage due to metabolic alterations [25]. Among the metabolic risks are dyslipidemia, which occurs when the cholesterol, triglycerides, and low- and high-density lipoprotein values are affected negatively [21,24,26,27], and diabetes and insulin resistance syndrome, associated with a chronic inflammatory state [28], which causes a morbidity level higher than 65% in individuals with cardiovascular disease [29,30,31]. Hypertension, on the contrary, is the fundamental risk factor for cardiovascular damage [32]; in Mexico, the prevalence of hypertension is approximately 25% of the population [30].

### 1.2. Insomnia, Stress and Cardiovascular Disease 

Insomnia is the most common sleep disorder [33], and it has been related to several health problems, such as cardiovascular disease or stress [34,35]. This can be seen in the model of Levenson, Kay, and Buysse, where they explained how genetic vulnerability, along with other stressful events and predisposing risk factors, causes anomalies in neural biological processes; this causes neurophysiological hyperactivity and provokes physiological behavioral processes, resulting in insomnia and further health alterations [36]. In many cases, insomnia can become a chronic condition which may imply an increase in health costs due to work or school absenteeism [37].

In Mexico, a previous study found that 40% of the population had insomnia before the pandemic [38]; other studies have shown that even after one to three years of respiratory disease outbreaks, insomnia persists [39] which has been identified in China as well [40].

Along with insomnia, stress is a mediator of homeostasis, which modulates metabolism causing the fight-or-flight response in an organism [41,42] through a “massive shock” of the sympathetic nervous system, provoking an increase in arterial blood pressure, blood flow, metabolism, and the increase in coagulation cascades from a stressful situation [43]. However, if these conditions persist for a long period of time, they can cause a state of illness [44]. Thus, there is clearly a close relationship between stress and cardiovascular disease. When the arterial pressure is between 75 and 100 mm/Hg during a state of fear [43] or in an intense emotional stress state, a rupturing of an atherosclerotic plaque can occur [15]. This can be worsened by neurophysiological hyperactivation with insomnia [36], causing cardiovascular alterations as well. It has been reported that stress levels, in general, have increased as a result of the COVID-19 pandemic [7,8,9].

In China, 330 million people are diagnosed with cardiovascular disease, which is usually related to anxiety [45]. The association between insomnia and health problems is not commonly found in the literature, but recent research shows its connection with stress [46]. Sleep disruption, such as duration alterations, sleep apnea, or even insomnia, have been reported to be able to influence cardiovascular disease course and outcomes [47].

There is not sufficient evidence to show the association of stress, insomnia, and cardiovascular disease, which in some part has been well documented in health care professionals, scientists, or people with hypertension. In our paper, we show that besides the well-known risk factors for cardiovascular disease, insomnia and stress actively participate by increasing the risk in a sample of the Mexican general population, regardless of profession.

Therefore, the general objective of this research was to identify the impact of insomnia and stress on cardiovascular risk during the COVID-19 pandemic in the Mexican population. Specific objectives include improving the association of these conditions with mental and physical health and to understand their long-term consequences. 

## 2. Materials and Methods

### 2.1. Participants and Design

The study was conducted from June to July 2020; sample size was determined and participants were chosen using the non-probabilistic method. Participants (N = 395; 61% female) were aged between 18 and 72 years (median = 33.9 years), all of whom had been confined due to the COVID-19 pandemic. Participants received a request to participate through social media. They were from Mexico State, Queretaro, Yucatan, Campeche, Guanajuato, Aguascalientes, Puebla, Tamaulipas, Tabasco, San Luis Potosi, Quintana Roo, Veracruz, Sonora, Michoacan, Nuevo Leon, Oaxaca, Coahuila, Durango, Jalisco, Hidalgo, and Guerrero. The educational level of the participants was categorized into 61.3% college, 22.8% high school, and 12.7% postgraduate, with the remaining percentage referring to elementary and junior high school. The occupational status of the sample was divided into 64.8% students, 24.3% housewives, 6.6% unemployed, and 1.5% retired workers.

### 2.2. Measurements 

A questionnaire to evaluate cardiovascular risk included each of the following variables: gender (female = 1 and male = 2), age (<20 years old = 1, 21–30 years old = 2, 31–40 years old = 3, 41–50 years old = 4, and >50 years old = 5), inherited cardiovascular background (no = 1 and yes = 2), inherited metabolic background (no = 1 and yes = 2), body mass index (within normal boundaries = 1 and overweight = 2), smoking behavior (no = 1 and yes = 2), alcohol consumption (no = 1 and yes = 2), exercise (yes = 1 and no = 2), nutrition (healthy = 1 and unhealthy = 2), and high blood pressure, diabetes, hypercholesterolemia, hypertriglyceridemia, depression, and anxiety in the last two months (these six variables had values of never = 1, once = 2, twice = 3, three times = 4, more than three times = 5, and every day = 6). This information comprised the cardiovascular risk index.

The main analysis instrument used to identify stress was “The Personal Resource Instrument” by Palacios [48]; this instrument uses a Likert scaling method and is constituted by three factors for total stress measurement: fatigue, burnout, and coping. The fatigue factor includes 10 questions, which inquire about the presence of excessive tension, insomnia, headaches, constant fatigue, and tiredness at the end of the working day. The second factor asks about burnout, with six questions on burnout, feeling overwhelmed, exhaustion, and pressure. The last factor contains questions about dealing with stress; in other words, whether the person is capable of integrating stress-relief techniques in their daily life, such as exercise and alternative actions (e.g., taking it easy, facing stressful situations, or talking about their problems). For insomnia, the “seven-item guideline” was used to measure the severity of insomnia, written by Guillén and Santos in 2008. These guidelines comprise a brief questionnaire used to evaluate the level of severity of the day and night components of insomnia. The questionnaire is answered using a Likert scale from 0 to 4, where 0 refers to none and 4 refers to a lot. For the evaluation, a total score ranging from 0 to 28 is obtained. The rank setup for the proposed measurement is 0–7, no insomnia observed; 8–14, subclinical insomnia; 15–21, clinical insomnia; and 22–28, severe insomnia [49]. 

### 2.3. Procedure

A questionnaire was created in Google Forms to facilitate participation during the confinement, based on the Official Mexican Standards and Regulations for patients’ clinical records, NOM-004-SSA3-2012, to collect medical histories, including the comorbidities, cardiovascular risk factors, and sociodemographic variables of the Mexican population in general. A written consent form was provided to all participants, which explained in detail the purpose of the research and that their participation in the study was completely anonymous, confidential, and without risk to the participants; it had to be read completely and accepted by the participant prior to being able to enter into the study and respond to the instruments.

### 2.4. Data Analysis 

For the statistical analysis, SPSS (Statistical Package for the Social Sciences) was used. The data were entered and analyzed; descriptive statistics were used for the variables, with measures of central tendency and dispersion, including cardiovascular risk, sociodemographics, and the severity of insomnia and stress. The Shapiro–Wilk Test was used to test the normal distribution. Because insomnia (S-W = 0.96, *p* < 0.001) and stress (S-W = 0.98, *p* < 0.001] had non-normal distribution, non-parametric tests were used. Mann–Whitney U tests were used to determine differences for insomnia and stress between cardiovascular risk factors. Spearman Rho’s correlation was used to assess the relationship between cardiovascular risk, insomnia, and stress. We used one-way ANOVA to compare the statistical means of the cardiovascular risk for the insomnia and stress groups. When the results were obtained, and in order to find the statistically significant differences, we used a post-hoc test with the Scheffé method. Simple linear regression was used to examine the relationship among cardiovascular risk and levels of insomnia and stress, the value *p* < 0.05 was accepted as statistically significant.

## 3. Results

### 3.1. Characteristics of Cardiovascular Risk Profile in Participants

In order to distinguish the factors, along with insomnia and stress, that could possibly contribute to cardiovascular risk, Table 1 and Table 2 were built. Table 1 shows increased insomnia in women, in individuals who do not exercise often, with a cardiovascular history, unhealthy nutrition, hypercholesterolemia, depression, and anxiety.

Table 2 shows that the risk profile of the participants revealed a tendency of high stress levels in women, with differences in the participants who were overweight, had unhealthy nutritional status, were depressed, or were anxious.

### 3.2. Description of the Cardiovascular Risk Groups

For the index of cardiovascular risk, we obtained the sum for each variable, with a minimum value of 15 and a maximum of 43. The median was 22.39, and the mode was 21, with a standard deviation of 3.599. The frequency distribution to identify the cardiovascular risk groups showed that 21.3% had a low cardiovascular risk, 25.6% had a moderate cardiovascular risk, 28.1% had a high cardiovascular risk, and 25.1% had a very high cardiovascular risk. Figure 1 shows the differences between gender and cardiovascular risk group.

### 3.3. Cardiovascular Risk Groups with Insomnia and Stress Comparison

In order to compare the stress and insomnia groups, we used one-way ANOVA; the statistical means of the severe insomnia and the stress indexes are represented in Figure 2, along with fatigue, burnout, and coping for the cardiovascular risk groups shown in Figure 3.

The one-way ANOVA showed that, among the cardiovascular risk groups, the severe insomnia index variable had a statistically significant relationship (*F* = 36.093, *p* = 0.000) with stress (*F* = 20.091, *p* = 0.000). All CVR groups presented with stress. When comparing CVR groups with respect to fatigue, burnout, and coping with stress, the high and very high CVR groups showed increased fatigue and decreased coping. In the very high CVR group the following statistics were calculated: fatigue (*F* = 40.967, *p* = 0.000), burnout (*F* = 35.218, *p* = 0.000), and coping with stress (*F* = 9.375, *p* = 0.000).

After finding statistical significance with the one-way ANOVA, we used a post-hoc test with the Scheffé method on the different variables of insomnia, stress, and the three subsections (fatigue, burnout, and stress coping), alongside cardiovascular risk. We considered the low cardiovascular risk group as the control group for this analysis. Figure 4 and Figure 5 represent the variable comparison with the control group.

In Figure 4, it can be observed that when insomnia was compared to the control group (the low cardiovascular risk group), there was a statistically significant difference between the high cardiovascular risk group (*p* = 0.001) and the very high cardiovascular risk (*p* = 0.000) group.

In Figure 5, it can be observed that in comparison to the control group, the low cardiovascular risk group, with respect to fatigue, burnout, and coping variables, a statistically significant difference was found between the control group and the high cardiovascular risk group (*p* < 0.01) as well as with the very high cardiovascular risk group (*p* < 0.01). 

### 3.4. Predictors of Cardiovascular Risk Index

Spearman correlations were conducted to examine relationships between cardiovascular risk, insomnia, and stress. The data showed a significant, moderate positive correlation between cardiovascular risk and insomnia (r = 0.485; *p* < 0.001) and stress (r = 0.482; *p* < 0.001); furthermore, we found that at higher levels of stress, insomnia in the participants increased (r = 0.549; *p* < 0.001).

Finally, we examined the influence of insomnia and the three subscales of stress on the cardiovascular risk indicator. A stepwise multiple regression analysis was performed considering all of the cardiovascular risk indicators as the dependent variables. The model was statistically significant. Table 3 provides details of the regression coefficients in the linear regression model.

The regression analysis showed two independent and significant predictors that entered the regression model. In the first step, the fatigue stress subscale was included (F = 211.901, *p* < 0.001) as the main predictor. In the second step, insomnia was incorporated into the regression equation (F = 111.540, *p* < 0.001). Fatigue and insomnia were associated with an increase in cardiovascular risk in this sample, with an R^2^ Nagelkerke of 363. 

Collinearity was verified, exhibiting medium tolerance values. The value of the variance inflation factor did not exceed 4, which indicates the absence of multicollinearity between the independent variables in the regression model; that is, the proportions of the decomposition of the variance did not overlap because collinearity was not found.

## 4. Discussion

Cardiovascular risk factors are the main cause of COVID-19 morbidity and mortality. Moreover, in the pandemic, confinement and virtuality changed the lifestyle, and due to uncertainty and the number of deaths worldwide, a major incidence of sleep disorders and cases of stress emerged; these two in combination with cardiovascular risk increased COVID-19 morbidity and mortality.

Insomnia has affected more people during the COVID-19 pandemic, due to having to face higher levels of stress as a result of health concerns, economic distress, and changes in social life and health routines [35]. In the present study, insomnia was identified in the sample population as follows: 44.1% of participants had no insomnia, while the rest had subclinical, moderate clinical, and severe clinical insomnia—38%, 14.2%, and 3.8%, respectively, with females representing 59% of the total. These values are higher than those found by Pappa et al. [23], where the prevalence of insomnia was 34%. In Huaracaya’s study [40], the percentage observed was the same as in Pappa et al.’s [23] study, while Fu et al. [5] observed 30%. However, all of these studies identified that females were more affected than males, coinciding with the findings in the present research.

In addition, the present research results showed that only 22.8% of the participants had no stress, while the remaining 77.2% presented the following levels of stress: low, moderate, and severe at 27.1%, 25.1%, and 25%, respectively. These results are the highest reported. In Taylor et al.’s study [2], 54% of the participants reported stress, while Wang et al. [3] reported moderate-to-severe stress in 8.1% of their sample population. These differences may be due to the use of different parameter measurements and, more importantly, the stage of the pandemic at which the studies were performed; lower percentages were present in the early stages. In contrast, it has been observed in some studies that females are more affected by stress than males [7,9].

Patients with cardiovascular disease have been identified as being severely affected by COVID-19, with a 12-fold increase in the risk of mortality [14]. Insomnia [8] and stress [44] have also been increasing due to COVID-19, and they are considered relevant risk factors, as observed in the data obtained in this research. Another strength of our study was that after analyzing the one-way ANOVA results, for the cardiovascular risk groups related to all of the variables, we found statistically significant differences for insomnia and stress. The means of insomnia and stress increased when the cardiovascular risk was high or very high. The coping with stress variable was not sufficient to decrease the risk in the very high cardiovascular risk group. Considering the low cardiovascular risk group as the control group, it was found that insomnia, fatigue (stress), and burnout (stress) were significantly different in the high and very high cardiovascular risk groups; meanwhile, in terms of total stress and coping with stress, there was only a significant difference for the very high cardiovascular risk group, and still coping with stress could not reduce the risk. 

One of the weaknesses of the present research is that the study was based on the information obtained from an online questionnaire, due to the confinement; therefore, some participants may have not reflected accurately the severity of their stress levels or insomnia information. We understand the need for an increase in the number of participants to be able to suggest that the cardiovascular risk present in the sample population will increase with the presence of stress and insomnia. We consider that the use of validated instruments allowed us to find the association between the variables on cardiovascular risk factors.

Finally, this research showed that only a fourth of the sample population presented low cardiovascular risk; meanwhile, the rest of the sample had either moderate or severe cardiovascular risk, revealing the importance of the impact of cardiovascular diseases in Mexico. Another important piece of evidence found in this study was that more than half of the sample population presented some degree of insomnia and stress, affecting females more than males. Therefore, this research provides a better understanding of the reality of the COVID-19 confinement, as these data were higher than those described before the pandemic [50]. The data obtained in this study can help to direct appropriate preventive information to the population in general; because, not only during the pandemic, but after it had been declared to end, cardiovascular disease is the number one cause of death and still is the main reason for the morbidity and mortality of COVID-19 patients, and it is closely related to stress and insomnia [29]. 

## 5. Conclusions

The present study allows us to conclude that the known cardiovascular risk factors, such as gender, age, body mass index, tobacco use, depression, and anxiety, showed an impact on cardiovascular risk during the COVID-19 pandemic along with the presence of insomnia and stress, on a sample of the Mexican population. It is important to emphasize that the findings of cardiovascular risk increasing with insomnia and stress can enable better prevention; this information, complemented by the use of medical history, can help patients to understand their health risks better along with the use of valid and trustworthy information.

This investigation considers the importance of the multidisciplinary treatment by health professionals regarding insomnia and stress, not only on the biological side, but on the psychological side and its relationship with the environment, prescribing the use of relaxation techniques, better sleep patterns, exercise, nourishment, and implementing recreational activities to improve patients’ general health. Coping strategies, prevention campaigns, and patient education should be developed alongside health professionals, in order to support people’s mental and physical health. In this way, the present investigation provides evidence that supports the evaluation of the impact of stress and insomnia on CVR to be able to offer better quality of life for patients.

Finally, this study presents a better vision of how to analyze the repercussions of lifestyle alterations, such as confinement, on the mental and cardiovascular health of individuals. Future studies should consider a larger sample population, in which additional data could be obtained to understand the connection between mental health disorders, such as chronic stress, anxiety, depression, and mood disorders, and heart and metabolic diseases, in order to implement safe and effective intervention strategies.

## Figures and Tables

**Figure 1 ijerph-20-07135-f001:**
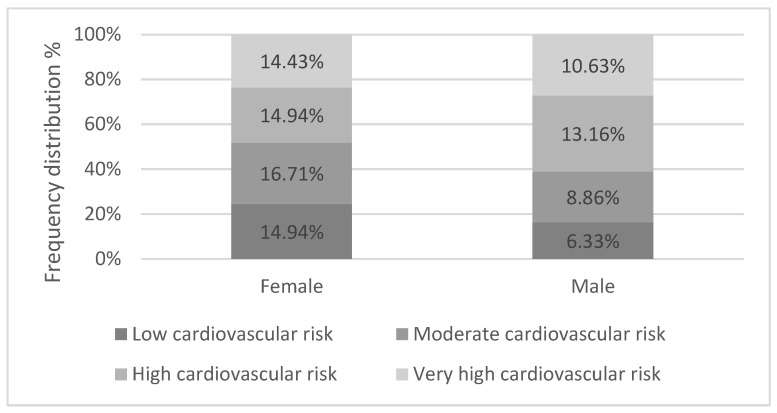
Percentage of responders (n = 395) in each cardiovascular risk category according to gender.

**Figure 2 ijerph-20-07135-f002:**
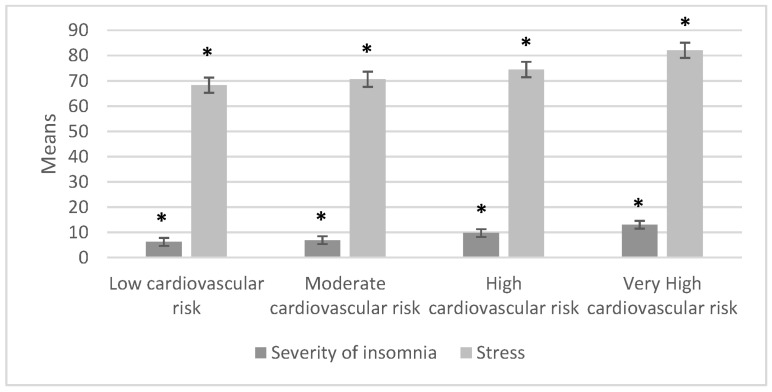
Comparison of the means of the severity of insomnia and stress indexes within the cardiovascular risk groups (n = 395). * These variables showed statistically significant differences.

**Figure 3 ijerph-20-07135-f003:**
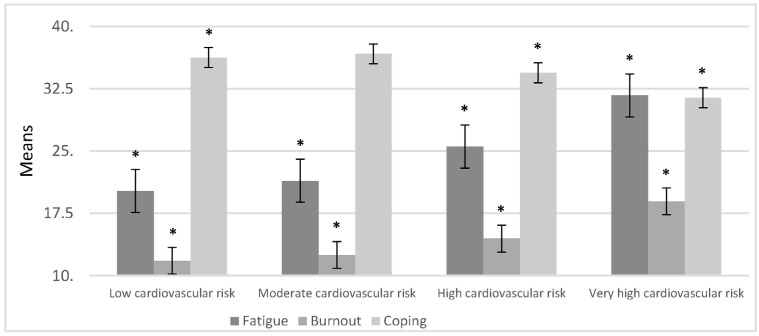
Comparison of the means of cardiovascular risk with fatigue, burnout, and coping with stress (n = 395). * These variables showed statistically significant differences.

**Figure 4 ijerph-20-07135-f004:**
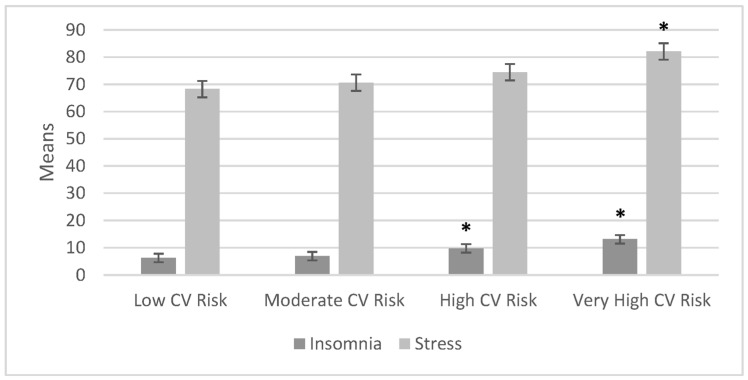
Comparisons with the control group: regarding low cardiovascular risk, with insomnia, a * statistically significant difference was found for the high cardiovascular risk group and with the very high cardiovascular risk group. When the control group was compared to the other groups with respect to stress, a * statistically significant difference was found with the very high cardiovascular risk group (n = 395).

**Figure 5 ijerph-20-07135-f005:**
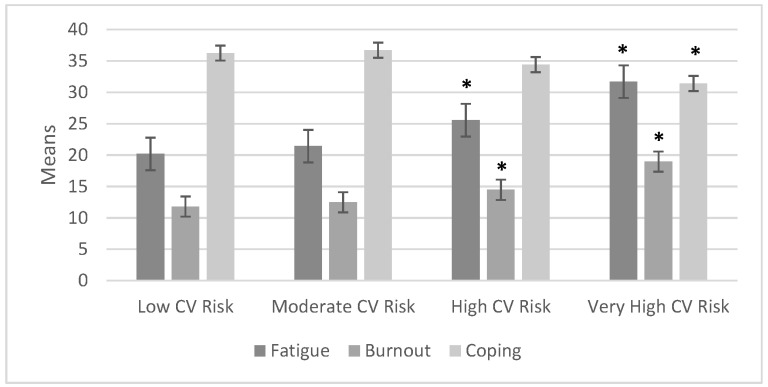
Comparison to the control group, the low cardiovascular risk group, with respect to fatigue, burnout, and coping variables. Where there is an *, there is a statistically significant difference with the high-risk group and the very high-risk group (n = 395).

**Table 1 ijerph-20-07135-t001:** Differences in insomnia levels between cardiovascular risk factors (N = 395).

Factors	%	M	SD	U	*p*	d
1. Gender						
Male	39	8.16	5.2	21,199.00	0.017	0.27
Female	61	9.70	5.9			
2. Age						
<20 years old	9	10.17	4.7	7283.50	0.127	0.20
>41–50 years old	91	9.00	5.7			
3. Body mass index						
Normal weight	44	9.33	5.8	20,084.50	0.515	0.07
Overweight	22	8.91	5.6			
4. Smokes						
Yes	13	9.98	6.3	9508.00	0.333	0.17
No	87	8.97	5.6			
5. Drinks alcohol						
Yes	75	9.06	5.7	14,844.00	0.845	0.03
No	25	9.24	5.7			
6. Exercises						
Yes	74	8.68	5.3	13,138.00	0.056	0.28
No	26	10.29	6.6			
7. Inherited metabolic background						
Yes	75	9.39	5.8	16,117.00	0.110	0.20
No	25	8.22	5.3			
8. Inherited cardiovascular background						
Yes	74	9.538	5.9	17,296.00	0.023	0.29
No	26	7.883	4.7			
9. Nutrition						
Healthy	68	8.19	5.6	11,647.00	0.001	0.50
Unhealthy	22	11.02	5.4			
10. High blood pressure						
Yes	9	10.417	5.9	5443.50	0.124	0.26
No	91	8.933	5.6			
11. Diabetes						
Yes	3	9.61	4.7	2199.00	0.492	0.09
No	97%	9.08	5.7			
12. Hypercholesterolemia						
Yes	6	10.41	5.9	2936.00	0.005	0.55
No	94	8.93	5.6			
13. Hypertriglyceridemia						
Yes	6	10.08	4.9	3587.500	0.206	0.19
No	94	9.00	5.7			
14. Depression						
Yes	43	11.91	5.5	8017.50	0.001	1.09
No	57	6.54	4.3			
15. Anxiety						
Yes	63	10.16	5.4	8107.50	0.001	0.87
No	37	5.82	4.1			

Note: % = percentage, M = mean, SD = Standard deviation, U = Mann whitney test, *p* = *p* value, d = effect size Cohen’s d.

**Table 2 ijerph-20-07135-t002:** Differences in the stress levels between cardiovascular risk factors (N = 395).

Factors	%	M	SD	U	*p*	d
1. Gender						
Male	39	69.68	14.2	24,285.00	0.001	0.52
Female	61	76.85	13.2			
2. Age						
<20 years old	9	77.74	17.4	6949.500	0.321	0.28
>41–50 years old	91	73.70	13.6			
3. Body mass index						
Normal weight	44	71.64	14.0	23,112.50	0.001	0.38
Overweight	22	76.93	13.5			
4. Smokes						
Yes	13	74.13	14.6	8977.50	0.788	0.00
No	87	74.04	13.9			
5. Drinks alcohol						
Yes	75	73.93	14.1	15,024.00	0.705	0.03
No	25	74.41	13.7			
6. Exercises						
Yes	74	73.46	13.8	13,541.00	0.133	0.16
No	26	75.74	14.4			
7. Inherited metabolic background						
Yes	75	74.26	13.8	14,845.50	0.766	0.06
No	25	73.41	14.9			
8. Inherited cardiovascular background						
Yes	74	74.56	13.8	16,130.00	0.273	0.13
No	26	72.62	14.5			
9. Nutrition						
Healthy	68	73.14	13.8	14,588.50	0.022	0.20
Unhealthy	22	75.97	14.3			
10. High blood pressure						
Yes	9	75.77	12.7	5900.50	0.404	0.14
No	91	73.81	14.1			
11. Diabetes						
Yes	3	74.19	14.1	2812.50	0.406	0.25
No	97	70.61	12.0			
12. Hypercholesterolemia						
Yes	6	75.77	12.7	3682.50	0.156	0.33
No	94	73.81	14.1			
13. Hypertriglyceridemia						
Yes	6	74.16	13.9	4771.50	0.341	0.20
No	94	71.26	14.6			
14. Depression						
Yes	43	80.35	13.1	8969.00	0.001	0.96
No	57	68.27	11.9			
15. Anxiety						
Yes	63	76.6	12.7	8961.50	0.001	0.86
No	37	65.9	11.7			

Note: % = percentage, M = mean, SD = Standard deviation, U = Mann whitney test, *p* = *p* value, d = effect size Cohen’s d.

**Table 3 ijerph-20-07135-t003:** Multiple regression analysis to predict cardiovascular risk with stress (fatigue) and insomnia (N = 395).

Model	b	SE	β	CI = 95%	Adj R2	Collinearity Statistics	
						Tolerance	VIF
Fatigue	0.247	0.029	0.481 ***	0.190–0.305	0.34	0.500	2.001
Insomnia	0.128	0.046	0.157 **	0.037–0.219	0.35	0.500	2.001

Note: b = beta, SE = Standard error, β = beta standardized, *p* = *p* value, CI = confidence interval. R adj = R2 adjusted. VIF = value of the variance inflation factor. ** *p* < 0.01, *** *p* < 0.001.

## Data Availability

Participant’s data were kept confidential and the questionnaire was anonymous. The information obtained in this investigation will not be used for purposes different from research and the present investigation followed the Official Mexican Standards and Regulations for patients’ clinical records, NOM-004-SSA3-2012, to maintain confidentiality.

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
