# Peer review of "The Association of Insomnia and Stress on Cardiovascular Risk Factors during COVID-19 Confinement in the Mexican Population"

_ijerph, 2023, doi:10.3390/ijerph20237135_

Round 1

Reviewer 1 Report (Previous Reviewer 1)

Comments and Suggestions for Authors

This paper presents an interesting addition to existing literature. However, there are several areas where improvements could be made to enhance clarity, precision, and overall quality of the study. Here are my suggestions (not in any specific order):

Clarification on Causal Relationships: The methods employed in this study do not permit the establishment of causal relationships. Therefore, the term “impact” should be substituted with “association” to more accurately reflect the nature of the relationships being studied.

Condense the Introduction: The introduction, particularly the sections before/preceding subsection 1.1, could be significantly shortened. Aim to maintain brevity while ensuring that all necessary context and background information are provided.

Revise the Statement of Objectives: The sentence “The general objective of this research was to identify the impact of insomnia and stress on cardiovascular risk during the COVID-19 pandemic in the Mexican population, to enhance understanding of this condition and its relationship with mental and physical health, and to discern its long-term consequences, as well as to develop interventions to prevent future public health problems,” is lengthy and somewhat ambiguous. Clarify whether there are specific objectives in addition to the general objective, and consider breaking this statement into shorter, more concise sentences. Also, revise or remove any objectives that were not directly addressed in the study, such as “to develop interventions…”

Provide More Details on Participant Recruitment: The paper should offer a more comprehensive description of how participants were recruited for the study.

Rephrase the Statement on Linear Regression: The statement “Simple linear regression was used to demonstrate significant impact on index of cardiovascular risk based” should be revised. Instead of implying that regression analysis can “demonstrate/prove” a relationship, it should be stated that this method was used to “examine” or “investigate” the relationship.

Clarify Table Column Names: Ensure that all column names in the tables are fully spelled out and clearly defined to prevent any potential confusion for readers.

Discuss Strengths and Limitations in the Discussion Section: Instead of including the strengths and limitations of the study in the Conclusion, relocate this content to the Discussion section. This provides an opportunity to critically evaluate the study’s design, execution, and findings.

Shorten and Refine the Conclusion: Aim for a concise and focused conclusion that summarizes the main findings and their implications. Consider moving some of the current content in the Conclusion to the Discussion section.

Comments on the Quality of English Language

I think that engaging a professional editor will improve the quality of the paper.

Author Response

REVIEWER 1

This paper presents an interesting addition to existing literature. However, there are several areas where improvements could be made to enhance clarity, precision, and overall quality of the study. Here are my suggestions (not in any specific order):

  • Clarification on Causal Relationships: The methods employed in this study do not permit the establishment of causal relationships. Therefore, the term “impact” should be substituted with “association” to more accurately reflect the nature of the relationships being studied.

This change has been made in the title

  • Condense the Introduction: The introduction, particularly the sections before/preceding subsection 1.1, could be significantly shortened. Aim to maintain brevity while ensuring that all necessary context and background information are provided.

Section 1.1 was shortened.

  • Revise the Statement of Objectives: The sentence “The general objective of this research was to identify the impact of insomnia and stress on cardiovascular risk during the COVID-19 pandemic in the Mexican population, to enhance understanding of this condition and its relationship with mental and physical health, and to discern its long-term consequences, as well as to develop interventions to prevent future public health problems,” is lengthy and somewhat ambiguous. Clarify whether there are specific objectives in addition to the general objective, and consider breaking this statement into shorter, more concise sentences. Also, revise or remove any objectives that were not directly addressed in the study, such as “to develop interventions…”

General and specific objectives were stated as suggested.

  • Provide More Details on Participant Recruitment: The paper should offer a more comprehensive description of how participants were recruited for the study.

Participants received invitation to participate through social media, which was a safe way to recruit participants during the confinement.

  • Rephrase the Statement on Linear Regression: The statement “Simple linear regression was used to demonstrate significant impact on index of cardiovascular risk based” should be revised. Instead of implying that regression analysis can “demonstrate/prove” a relationship, it should be stated that this method was used to “examine” or “investigate” the relationship.

The statement of linear regression was modified accordingly, as a method to examine the relationship among cardiovascular risk, stress and insomnia.

  • Clarify Table Column Names: Ensure that all column names in the tables are fully spelled out and clearly defined to prevent any potential confusion for readers.

Column names in the table are defined.

  • Discuss Strengths and Limitations in the Discussion Section: Instead of including the strengths and limitations of the study in the Conclusion, relocate this content to the Discussion section. This provides an opportunity to critically evaluate the study’s design, execution, and findings.

Relocation of strengths and limitations was done in the text as suggested.

  • Shorten and Refine the Conclusion: Aim for a concise and focused conclusion that summarizes the main findings and their implications. Consider moving some of the current content in the Conclusion to the Discussion section.

Conclusion was shortened and refined.

Reviewer 2 Report (Previous Reviewer 3)

Comments and Suggestions for Authors

sufficiently revised 

Accept 

Author Response

REVIWERS 2 and 3 had no comments.

Sufficiently revised

Reviewer 3 Report (Previous Reviewer 2)

Comments and Suggestions for Authors

I thank the authors for the explanation and addition of the required attributes as part of the corrections after the review. After a detailed study of the manuscript, I note its improvement and see no reason why the manuscript could not be published.

Author Response

REVIWERS 2 and 3 had no comments.

Reviewer 4 Report (New Reviewer)

Comments and Suggestions for Authors

Thank you for reviewing the paper titled “The impact of insomnia and stress on cardiovascular risk factors during confinement due to COVID-19 in the Mexican population.” The paper is organized, and well-written, but some questions raised into consideration:

Title: I recommend writing the type of study at the end of the title.

ABSTRACT: very good, but the method needs to be mentioned clearly.

Introduction: Too long introduction. The text could be shortened. It need more example of studies about insomnia and cardiovascular risk which is similar or against this one.

 Mention the validity and reliability of the questionnaire.

ANOVA, please mention one or two-way ANOVA.

 “Sample size was calculated” Where is sample size calculation?

“This instrument uses a Likert scaling method and is constituted by three factors for total stress measurement: Fatigue, burnout, and coping”. To what exactly is the author referring?

Mention inclusion and exclusion criteria for the study, please

Discussion: unfortunately, this does not cover all results

Conclusions: must be concise and informative, please rewrite briefly.

 Some sentences of the conclusion should be better addressed in the discussion section.

Check the manuscript's English language and grammar

Check references well

 Overall, this manuscript is written exceptionally well.

Comments on the Quality of English Language

Check the manuscript's English language and grammar

Author Response

REVIEWER 4

Thank you for reviewing the paper titled “The impact of insomnia and stress on cardiovascular risk factors during confinement due to COVID-19 in the Mexican population.” The paper is organized, and well-written, but some questions raised into consideration:

  • Title: I recommend writing the type of study at the end of the title.

The title had a minor modification regarding a more appropriate description of the research, we hope your approval on the change.

  • ABSTRACT: very good, but the method needs to be mentioned clearly.

The main method of how participants data were obtained was included in the abstract.

  • Introduction: Too long introduction. The text could be shortened. It need more example of studies about insomnia and cardiovascular risk which is similar or against this one.

Introduction has been shortened.

References 1 to 3 below, are examples of lack of sufficient evidence to show the association of stress, insomnia and cardiovascular disease, which in some part has been well documented in health care professionals, scientists or people with hypertension. In our paper, we show that besides, the well-known risk factors for cardiovascular disease, insomnia and stress actively participate by increasing the risk in a sample of Mexican general population, despite of profession. We included part of this information in the introduction as suggested.

In China, 330 million people are diagnosed with cardiovascular disease, which is usually related to anxiety (1).

The association between insomnia and health problems is not commonly found in the literature, but recent research shows its connection with stress (2).

Sleep disruption such as duration alterations, sleep apnea or even insomnia, have been investigated to be able to influence cardiovascular disease course and outcomes (3).

  1. Zhang L, Bao Y, Li G, Tao S, Liu M. Prevalence and risk factors of cardiovascular diseases and psychological distress among female scientists and technicians. J Zhejiang Univ Sci B. 2022 Dec 15;23(12):1057-1064. doi: 10.1631/jzus.B2200162. PMID: 36518057; PMCID: PMC9758715.
  2. Fernandez-Mendoza J, Vgontzas AN. Insomnia and its impact on physical and mental health. Curr Psychiatry Rep. 2013 Dec;15(12):418. doi: 10.1007/s11920-013-0418-8. PMID: 24189774; PMCID: PMC3972485.
  3. Hall MH, Brindle RC, Buysse DJ. Sleep and cardiovascular disease: Emerging opportunities for psychology. Am Psychol. 2018 Nov;73(8):994-1006. doi: 10.1037/amp0000362. PMID: 30394778; PMCID: PMC6220679.
  • Mention the validity and reliability of the questionnaire.

The questionnaire was built using validated instruments and clinical information was obtained in compliance with the Official Mexican Standards and Regulations for the patients clinical records NOM-004-SSA3-2012, as mentioned in the method section.

  • ANOVA, please mention one or two-way ANOVA.

One-way ANOVA was used.

  • “Sample size was calculated” Where is sample size calculation?

Sample size was calculated and participants were chosen using the non-probabilistic method, participants (N = 395) as follows:

n= (Z2 * p * q) /e2

n= (1.9622 * 0.5 * 0.5) /0.052

n= 384.9 = 385

  • “This instrument uses a Likert scaling method and is constituted by three factors for total stress measurement: Fatigue, burnout, and coping”. To what exactly is the author referring?

We used an instrument which includes three factors for total stress measurement: Fatigue, burnout, and coping. The fatigue factor includes 10 questions, thatg inquire about the presence of excessive tension, insomnia, headaches, constant fatigue, and tiredness at the end of the working day. The second factor asks about burnout, with six questions on burnout, feeling overwhelmed, exhaustion, and pressure. The last factor contains questions about dealing with stress; in other words, whether the person is capable of integrating stress-relief techniques in their daily life, such as exercise and alternative actions (e.g., taking it easy, facing stressful situations, or talking about their problems). The above information in included in the methods section.

  • Mention inclusion and exclusion criteria for the study, please

The inclusion criteria were:

Mexican living in the national territory, between the ages of 18 and 72 and willing to participate in the study by signing the informed consent.

The exclusion criteria were:

Not being in the age window of 18 and 72.

Not willing to participate or sign the informed consent.

Not answering the questions accordingly.

  • Discussion: unfortunately, this does not cover all results

Discussion was revised in order to include all reported results.

  • Conclusions: must be concise and informative, please rewrite briefly.

Conclusions were reviewed.

  • Some sentences of the conclusion should be better addressed in the discussion section.

Some conclusion sentences were moved and addressed in the discussion.

  • Check the manuscript's English language and grammar

English was reviewed

  • Check references well:

We checked the references

  • Overall, this manuscript is written exceptionally well.

Round 2

Reviewer 1 Report (Previous Reviewer 1)

Comments and Suggestions for Authors

The manuscript has improve, however, there are still issue to be addressed. 

I suggest the authors rewrite the abstract to make it more factual and consistent with the title and methods used in the paper. Currently it states “The objective of this research was to identify the impact of insomnia and stress on cardiovascular risk.. ”, however, the title of the paper (which is consistent with the methods used) states “The association…”. I suggest to include the main quantitative results in abstract.

I think that in Introduction this paragraph “There is not sufficient evidence…” should follow after this paragraph “In China, 330 million people…”.

I think that the paper must do a much better job at explaining the why specific methods have been used and what are potential concerns of the methods employed.

Tables need more work. All tables should include N = sample size. In particular, Table 3 should include the sample size, adjusted R-squared, p-values to the null hypothesis that marginal effects are statistically significantly different from 0. In all tables please use standard notations for estimators such as mean, standard deviation and others.

This statement cannot be true “These results prove the study hypothesis that the cardiovascular risk present in the sample population increased with the presence of stress and insomnia. because statistical evidence is not a method to prove statements. Please rewrite. I do not see a comprehensive discussion about the weaknesses of the study. This must be addressed.

Comments on the Quality of English Language

The quality of English should be improved. 

Author Response

The manuscript has improved, however, there are still issues to be addressed.  

I suggest the authors rewrite the abstract to make it more factual and consistent with the title and methods used in the paper. Currently it states “The objective of this research was to identify the impact of insomnia and stress on cardiovascular risk”, however, the title of the paper (which is consistent with the methods used) states “The association…”. I suggest to include the main quantitative results in abstract.

Thank you for the correction about the word change from impact to association and to include it in the abstract.

Abstract has been reviewed.

I think that in Introduction this paragraph “There is not sufficient evidence…” should follow after this paragraph “In China, 330 million people…”.

The order of the paragraphs has been changed.

I think that the paper must do a much better job at explaining the why specific methods have been used and what are potential concerns of the methods employed.

The present research was based on the participants to accept an informed consent and to fill out an online questionnaire due to the confinement, some potential concerns of the methods employed, may suggest that some participants did not reflect accurately the severity of their stress levels or insomnia information. That is why we understand that we need to increase the number of participants. We consider that the use of validated instruments allowed us to find the association between the variables on cardiovascular risk factors.

Tables need more work. All tables should include N = sample size. In particular, Table 3 should include the sample size, adjusted R-squared, p-values to the null hypothesis that marginal effects are statistically significantly different from 0. In all tables please use standard notations for estimators such as mean, standard deviation and others.

 Tables have been modified according to reviewer’s comments.

This statement cannot be true “These results prove the study hypothesis that the cardiovascular risk present in the sample population increased with the presence of stress and insomnia.”  because statistical evidence is not a method to prove statements. Please rewrite. I do not see a comprehensive discussion about the weaknesses of the study. This must be addressed.

The statement “These results prove the study hypothesis that the cardiovascular risk present in the sample population increased with the presence of stress and insomnia”, was corrected accordingly.

The weaknesses of the study were addressed.

This manuscript is a resubmission of an earlier submission. The following is a list of the peer review reports and author responses from that submission.

Round 1

Reviewer 1 Report

Comments and Suggestions for Authors

This manuscript examines an interesting research question. However, I feel that this manuscript has too many issues to be addressed.

Abstract Clarity: The Abstract contains a confusing statement with the mention of "from medical histories focused on cardiovascular risk, according to the NOM-004-SSA3-2012."

Missing Information in the Abstract: The Abstract lacks essential factual details regarding the research findings. This omission leaves the reader without a concise understanding of the study's main results, which is a crucial component of any scientific paper. Including a brief summary of the key results would provide the reader with a more comprehensive overview.

Methods Section: The Methods section is currently lacking in specificity and requires a more detailed description of the procedures and the reason.

Tables and Supplementary Material: The manuscript's tables, along with any related notes, need further elaboration.

Comments on the Quality of English Language

The manuscript would benefit from professional editing.

Author Response

This manuscript examines an interesting research question. However, I feel that this manuscript has too many issues to be addressed.

The issues described by the reviewers will be carefully addressed.

Abstract Clarity: The Abstract contains a confusing statement with the mention of "from medical histories focused on cardiovascular risk, according to the NOM-004-SSA3-2012."

In Mexico, physicians and health care professionals must comply to the Official Mexican Standards and Regulations for the patients clinical records NOM-004-SSA3-2012 in order to get medical information from patients or participants in a research study to maintain confidentiality. In the present investigation this regulation was followed and the information retrieved was kept confidential and used only for research purposes.

Missing Information in the Abstract: The Abstract lacks essential factual details regarding the research findings. This omission leaves the reader without a concise understanding of the study's main results, which is a crucial component of any scientific paper. Including a brief summary of the key results would provide the reader with a more comprehensive overview.

Essential factual details regarding the research findings have been included to the abstract. The main findings of the investigation demonstrate that women are more susceptible to stress and cardiovascular risk (CVR). However, stress is a major indicator of CVR than insomnia, but in the high and very high CVR groups, insomnia contributes along with stress; coping strategies rescue from the risk in the high CVR group, but not as expected to rescue in the very high CVR group. These findings suggest that sleep patterns and mental health alterations present during the pandemic persist even when the pandemic was declared to end and may contribute to the increase in cardiovascular risk and death in the long-term.

Methods Section: The Methods section is currently lacking in specificity and requires a more detailed description of the procedures and the reason.

The present investigation methods consisted in the following steps: first in choosing the sample size of individuals using the probabilistic method, resulting in 385 participants. Second, 395 confined people due to the pandemic, agreed to be participants and signed an informed consent. Then, they received a Google forms questionnaire containing the instruments of the study, which was conducted from June to July 2020. Finally, the data obtained were statistically analyzed with non-parametric tests.

Tables and Supplementary Material: The manuscript's tables, along with any related notes, need further elaboration.

Tables were changed as well as their titles, non-parametric tests were performed.

Reviewer 2 Report

Comments and Suggestions for Authors

Dear editor, thank you for the opportunity to carry out a review for International Journal of Environmental Research and Public Health. There are no any conflicts of interest of my own that might be perceived to have influenced my review.

In the manuscript entitled “The impact of insomnia and stress on cardiovascular risk factors during confinement due to COVID-19 in the Mexican population!, the authors address the issue of the consequences of the pandemic period (COVID-19), such as insomnia and stress in relation to cardiovascular diseases. The authors put three variables, which are significantly related to each other, into a causal relationship, which is confirmed by many studies published so far. However, in the context of the mentioned study, it is not clear what the cause is and what is the effect. Whether stress and insomnia are a consequence of the present risk of cardiovascular disease (and not only the risk), or vice versa. I recommend the authors to use more precise wording and not to draw general conclusions.

Abstract: I recommend presenting specific results and briefer methodological parts. The last sentence does not clearly correspond to the conclusion.

Results: In the case of tables 1 and 2, I recommend changing their titles. They do not correspond to the data stated in them, or they do not describe the essence of the problem being solved. At the same time, it is advisable to add the number of respondents to the individual answers, or their relative representation ("n"). It is advisable to include explanatory notes below the table.

Chapter 3.2: lines 170-179 are better listed in the methodology.

Figure 1: in my opinion, the title of the figure again does not convey the meaning of the content of the graph. I recommend replacing the word "relation" with e.g. with the word "distribution". This is the distribution of respondents according to gender and CVD risk. I suggest editing the sentence in lines 183-184.

Fig. 2 and 3: the "y" axis units are missing in the graphs.

Fig. 4 and 5: complete the titles of the graphs.

Discussion: lines 252-261 - this section summarizes the results/data that were not presented in the "Results" section. I suggest adding these missing data to the results section, but I recommend dividing the study group also by gender. Do the same with the stress issue discussed in lines 262-272.

Conclusion: the statement in lines 289-294 does not correspond to your findings and they are rather of a general nature. It is necessary to reformulate the sentence to make it clear what the authors' own findings are and what the general conclusions of other studies are.

Finally, I recommend giving the authors a chance to revise the manuscript after minor revision.

Author Response

Dear editor, thank you for the opportunity to carry out a review for International Journal of Environmental Research and Public Health. There are no any conflicts of interest of my own that might be perceived to have influenced my review.

In the manuscript entitled “The impact of insomnia and stress on cardiovascular risk factors during confinement due to COVID-19 in the Mexican population!, the authors address the issue of the consequences of the pandemic period (COVID-19), such as insomnia and stress in relation to cardiovascular diseases. The authors put three variables, which are significantly related to each other, into a causal relationship, which is confirmed by many studies published so far. However, in the context of the mentioned study, it is not clear what the cause is and what is the effect. Whether stress and insomnia are a consequence of the present risk of cardiovascular disease (and not only the risk), or vice versa. I recommend the authors to use more precise wording and not to draw general conclusions.

This research study was conducted during the pandemic (from June to July 2020), which caused mental health alterations in the population worldwide. The main findings demonstrated that women were more susceptible to stress and cardiovascular risk (CVR). However, stress is a major indicator of CVR than insomnia, but in the high and very high CVR groups, insomnia contributes along with stress; coping strategies rescue from the risk in the high CVR group, but not as expected to rescue in the very high CVR group. Insomnia and stress are related to the cardiovascular risk indicator, the influence of each one of them (insomnia and stress) towards the criterion (cardiovascular index) cannot be distinguished when they are combined with each other. Additionally, our results indicate that fatigue a total stress subscale, contributes to CVR.

Abstract: I recommend presenting specific results and briefer methodological parts. The last sentence does not clearly correspond to the conclusion.

The abstract has been reviewed and the main results of the study have been included.

Results: In the case of tables 1 and 2, I recommend changing their titles. They do not correspond to the data stated in them, or they do not describe the essence of the problem being solved. At the same time, it is advisable to add the number of respondents to the individual answers, or their relative representation ("n"). It is advisable to include explanatory notes below the table.

The titles of tables 1 and 2 were changed, the percentage of people for each risk factor and the respective notes in each table were added. 

Chapter 3.2: lines 170-179 are better listed in the methodology.

The information in lines 170-179 was moved to the methods section.

Figure 1: in my opinion, the title of the figure again does not convey the meaning of the content of the graph. I recommend replacing the word "relation" with e.g. with the word "distribution". This is the distribution of respondents according to gender and CVD risk.

The title of figure 1 has been changed as suggested.

I suggest editing the sentence in lines 183-184.

The sentence has been edited accordingly.

Fig. 2 and 3: the "y" axis units are missing in the graphs.

The Y axes represent in both figures the statistical means as it is stated in their title.

Fig. 4 and 5: complete the titles of the graphs.

The title has been included for figures 4 and 5 and they represent the variable comparison with the control group.

Discussion: lines 252-261 - this section summarizes the results/data that were not presented in the "Results" section. I suggest adding these missing data to the results section, but I recommend dividing the study group also by gender. Do the same with the stress issue discussed in lines 262-272.

The statistical analysis data was moved to the results section.

Conclusion: the statement in lines 289-294 does not correspond to your findings and they are rather of a general nature. It is necessary to reformulate the sentence to make it clear what the authors' own findings are and what the general conclusions of other studies are.

The author’s main findings have been included in the conclusion and contrasted with different reports.

Finally, I recommend giving the authors a chance to revise the manuscript after minor revision.

Reviewer 3 Report

Comments and Suggestions for Authors

I have read the study with great attention

The introduction is really good, it explains how the study is designed and give us a direction

simple collection was a bit complicated. I would recommend  to make a flowchart to make it simple.

I was wondering why author was using Google form instead of other important survey method’s 

Can you please explain why Mexican population have so much of insomnia

Can you please give some more light on some of the statistical test such as multicollinearty 

As that Study mentioned insomnia and fatigue are the major contributing factor. Can you please explain how insomnia and fatigue can be interrelated

Can there be a single entity which is missing?

How do you differentiate between the symptoms of different Barnold? Insomnia fatigue, depression anxiety when there can be able overlaps of symptoms?

Can you please explain whether you checked for all the other mental health issues such as depression, anxiety, psychosis mania, drug use overcrowding, excessive noises, violence and paranoia around area? 

Can you please explain if the factors where regularly distributed, or irregularly distributed
What is the nationality of Pearson test? Why not Mann Whitney test 

Conclusion also should provide guidance towards managing this problem

Author Response

I have read the study with great attention.

The introduction is really good, it explains how the study is designed and give us a direction simple collection was a bit complicated. I would recommend to make a flowchart to make it simple.

The order of the methods in this section has been changed to try to clarify them: first the sample size was calculated and participants were chosen according to the non-probabilistic method, then the participants, after signing an informed consent, completed the questionnaire containing the instruments to measure cardiovascular risk related to stress and insomnia during the pandemic.

I was wondering why author was using Google form instead of other important survey method’s 

Google forms was used to facilitate the access to the questionnaire to all participants, who were in confinement during the pandemic. This questionnaire included an informed consent and the measuring instruments to assess cardiovascular risk, stress and insomnia, as mentioned in the methods section.

Can you please explain why Mexican population have so much of insomnia

During the pandemic people around the world lived the confinement with long hours in front of a screen due to home office, home schooling or entertainment. Resulting in abnormal sleep schedules and tiredness, along with boredom, unhealthy nutrition and sedentary lifestyle, personal and pandemic stress all leading to insomnia. Unfortunately, this abnormal behaviors have continued in the post-pandemic, era (according to Hernández-Nava et.al.) for many people in the world, not only in Mexico, due to war, climate change and the uncertainty of future pandemics (Hernández-Nava RG, de la Luz Sánchez-Mundo M, García-Barrientos R, Espinosa-Solis V, Villalobos-Aguayo P, Salmerón-Muñiz NN, Anaya-Tacuba JD. Lifestyle Changes among Mexican People during the COVID-19 Lockdown in 2020: A Cross-Sectional Study. Healthcare (Basel). 2022 Dec 14;10(12):2537. doi: 10.3390/healthcare10122537. PMID: 36554061; PMCID: PMC9778622).

Can you please give some more light on some of the statistical test such as multicollinearity 

Collinearity was verified because insomnia and stress are related to the cardiovascular risk indicator data were incorporated and there is also an association between insomnia and stress in our data. Because of this, in the regression analysis it would be expected that the predictors of the model are related constituting a linear combination with cardiovascular risk (our data confirm this assumption). The above has the consequence that in the regression model using an enter method, the influence of each one of them (insomnia and stress) towards the criterion (cardiovascular index) cannot be distinguished when they are combined with each other, so it is not possible to give an explanation of the phenomenon under study. To solve this problem, we decided to 1) use a regression model with the stepwise method, instead of an enter method to report the independent effect of each variable, and 2) check the resulting collinearity in the regression model.

As that Study mentioned insomnia and fatigue are the major contributing factor. Can you please explain how insomnia and fatigue can be interrelated. Can there be a single entity which is missing?

Insomnia is associated to different factors, among these are abnormal sleep schedules, stress (personal, academic and at work) and poor nutrition (this research). Fatigue, according to Calabria, et.al., can be defined as physical and cognitive tiredness and is related to anxiety and depression. In our study, we measured total stress and fatigue, the fatigue factor included the presence of excessive tension, insomnia, headaches, constant tiredness at the end of the working day. We found that in people during the pandemic, the major predictable factor for cardiovascular risk was stress, but in the high and very high cardiovascular risk groups, insomnia contributed with stress, fatigue was used as a measure of total stress in our study.

(Calabria M, García-Sánchez C, Grunden N, Pons C, Arroyo JA, Gómez-Anson B, Estévez García MDC, Belvís R, Morollón N, Vera Igual J, Mur I, Pomar V, Domingo P. Post-COVID-19 fatigue: the contribution of cognitive and neuropsychiatric symptoms. J Neurol. 2022 Aug;269(8):3990-3999. doi: 10.1007/s00415-022-11141-8. Epub 2022 Apr 30. PMID: 35488918; PMCID: PMC9055007).

How do you differentiate between the symptoms of different Barnold (burnout). Insomnia fatigue, depression anxiety when there can be able overlaps of symptoms?

In our study, the entities stress, insomnia and fatigue were differentiated from anxiety and depression because, we used measuring instruments to retrieve the information from the participants. The Personal Resource Instrument by one of our co-authors, Dr. Palacios, is constituted by three factors for total stress measurement: Fatigue, burnout, and coping. For insomnia we used, the seven-item guideline, mentioned in the methods section. Anxiety and depression, were asked to the participants as part of the medical history format according to the Official Mexican Standards and Regulations for the patients clinical records NOM-004-SSA3-2012.

Can you please explain whether you checked for all the other mental health issues such as depression, anxiety, psychosis mania, drug use overcrowding, excessive noises, violence and paranoia around area? 

In this study, using the instruments mentioned in our method section, we measured, stress, insomnia, fatigue, burnout and coping and their relationship to cardiovascular risk. Anxiety and depression, were asked to the participants as part of the medical history format according to the Official Mexican Standards and Regulations for the patients clinical records NOM-004-SSA3-2012. We did not measure psychosis mania, drug use overcrowding, excessive noises, violence and paranoia, as these were not part of our main objective.

Can you please explain if the factors where regularly distributed, or irregularly distributed

Irregularly distributed

What is the nationality of Pearson test? Why not Mann Whitney test 

The normality of the variables was verified using the Shapiro–Wilk Test and non-parametric tests were used, the Rho coefficient of Spearman and Mann Whitney test as proposed by the reviewer.

Conclusion also should provide guidance towards managing this problem

The conclusion has been modified accordingly, and as we learned from the pandemics, obesity and diabetes increased the morbidity and mortality from COVID-19, therefore, future studies should consider a larger sample population, in which it could be obtained additional data to understand the connection between mental health disorders such as chronic stress, anxiety, depression, mood disorders with heart and metabolic diseases.